# Egg Allocation on *Anastrepha ludens* Larvae by Mass-Reared *Diachasmimorpha longicaudata* Females

**DOI:** 10.3390/insects16090926

**Published:** 2025-09-02

**Authors:** Jorge Cancino, Amanda Ayala, Erick Flores-Sarmiento, Flor de María Moreno, Lorena del Carmen Suárez, Sergio Marcelo Ovruski, Pablo Montoya

**Affiliations:** 1Facultad de Ciencias Agrícolas, UNACH, Entronque Carretera Costera y Pueblo de Huehuetán, Huehuetán 30660, CH, Mexico; 2Programa Operativo Moscas, DGSV-SENASICA-SADER, Camino a los Cacaoatales S/N., Metapa de Domínguez 30860, CH, Mexico; amanda.ayala.i@senasica.gob.mx (A.A.); erick.flores.i@senasica.gob.mx (E.F.-S.); flor.moreno.i@senasica.gob.mx (F.d.M.M.); 3Dirección de Sanidad Vegetal, Animal y Alimentos, Gobierno de la Provincia de San Juan, Nazario Benavides 8000 Oeste, Rivadavia, San Juan 5413, Argentina; lorenasuarez@conicet.gov.ar; 4Centros Científicos Tecnológicos, Concejo Nacional de Investigaciones Científicas y Técnicas CCT CONICET San Juan, Avenida Libertador General San Martín 1109, San Juan 5400, Argentina; 5Planta Piloto de Procesos Industriales Microbiológicos y Biotecnología (PROIMI-CONICET), Departamento de Control Biológico de Plagas, Avenida Belgrano y Pasaje Caseros, San Miguel de Tucumán 4001, Argentina; sovruski@conicet.gov.ar; 6Instituto de Biociencias, UNACH, Boulevard Príncipe Akishino S/N., Col. Solidaridad 2000, Tapachula 30798, CH, Mexico; pablo.montoya@unach.mx

**Keywords:** fruit fly parasitoid, parasitoid mass production, quality control parameters, female allocation, offspring fitness, reproductive strategy

## Abstract

**Simple Summary:**

The parasitoid *Diachasmimorpha longicaudata*, native to Southeast Asia, is often used as a fruit fly biocontrol agent in augmentative release programs worldwide. This parasitoid species is currently mass-produced and released in Mexico to control fruit fly pest species such as *Anastrepha ludens* and *A. obliqua.* Therefore, it is of the utmost importance to determine, under mass-rearing conditions, the optimal age at which *D. longicaudata* females can most efficiently oviposit on host larvae and produce high-quality adults, while reducing labor time and improving rearing productivity. In this study, we examined the egg allocation behavior of 5–10 day-old *D. longicaudata* females provided with *A. ludens* larvae as hosts under mass-rearing conditions. The following parameters were assessed for all female age groups: parasitized host mortality, parasitoid emergence, offspring sex ratio, and superparasitism. Superparasitism was based on both the number of scars on the puparium and of first-instar parasitoids inside the host. The results obtained provide valuable information for improving the mass production of *D. longicaudata*. The use of cages containing 5–7-day-old females may improve rearing quality because these females produce the highest female offspring sex ratio with improved fitness-related parameters. Furthermore, removing cages with older parasitoid females (≥8 days old) reduces production costs by minimizing the quantity of larval hosts and rearing commodities required.

**Abstract:**

The age-dependent reproductive behavior of parasitoid females under mass-rearing conditions may influence the quality of the biocontrol agents produced. Therefore, this study assessed the egg allocation behavior of *Diachasmimorpha longicaudata* (Ashmead) females at different ages under mass-rearing conditions. This parasitoid native to Southeast Asia is mass-reared on irradiated *Anastrepha ludens* (Loew) larvae at the Moscafrut Program facility in Mexico and is released in crop areas to suppress fruit fly pest species. Parasitoid females aged 5–10 days were used to assess quality control parameters, such as parasitized host mortality, parasitoid emergence, offspring sex ratio, and superparasitism. *Anastrepha ludens* puparia were checked and dissected 72 h after being exposed to female parasitoids to determine the number of scars on the puparium of first-instar parasitoids, respectively. Parasitized puparia were kept under lab conditions to assess adult emergence and offspring sex ratio. Host mortality, superparasitism, and parasitoid offspring sex ratio were affected by female age. The highest host mortality and superparasitism were caused by 5–7-day-old females, which also produced a significantly higher female-to-male offspring ratio with improved fitness-related parameters. The use of cages with females of these ages may improve the quality parameters of parasitoids and result in higher female-to-male offspring sex ratios. Furthermore, production costs could be lowered by reducing the quantity of hosts and rearing commodities. These findings are a useful contribution to improving the current method used for the mass rearing of *D. longicaudata* at the Moscafrut Program facility.

## 1. Introduction

Egg allocation by female parasitoids is mainly related to endogenous factors, such as egg load capacity, offspring sex allocation, and egg absorption, i.e., eggs are reabsorbed, mainly due to low host availability, allowing females to retrieve nutrients [1,2,3]. These factors are closely associated with the age of foraging females [3,4]. Exogenous factors, such as host availability, host–parasitoid ratio, host age, and host physiological condition, may also influence differential and selective egg allocation by female parasitoids [5,6,7]. Both endogenous and exogenous factors interact and play an important role in how female parasitoids allocate resources during egg production, which can be adjusted to improve both offspring fitness and sex ratio, thereby maximizing the reproductive process [2,5,8]. This usually represents a trade-off in resource investment between female reproduction and survival, which is mostly determined by the parasitoid’s physiological status, e.g., unlike mated parasitoid females, unmated females use resources to extend their survival [9]. Female age has been considered one of the most influential factors in the reproductive activity of parasitoids [1,3,10,11]. Parasitoid females also have the capacity to adjust the sex ratio of their offspring according to the quality of the host, and thus host selection maximizes the parasitoid’s reproductive capacity [5,12]. Different studies have focused on egg-laying behavior in relation to female age with the purpose of improving mass-rearing processes and release strategies of adults in the field for pest control. For instance, in some egg parasitoids of the genus *Trichogramma* Westwood, younger females have been reported to significantly allocate more resources to female eggs, which also had higher emergence rates [12,13]. Similarly, younger *Trichogramma pretiosum* (Riley) females exhibited the highest oviposition rates and produced female-biased offspring sex ratios [14,15]. A female-biased offspring sex ratio is also mostly produced by young females of the encirtid parasitoid *Aenasius bambawalei* (Hayat) [6]. However, physiological development and oviposition experience may influence both oviposition and parasitism rates in female parasitoids [11,16]. The augmentative parasitoid release is an effective technique for suppressing fruit fly populations [17], but its success may depend on efficient and cost-effective mass-rearing procedures, where the age of the female parasitoids used can be an important factor influencing the efficiency of mass production.

*Diachasmimorpha longicaudata* (Ashmead) (Hymenoptera: Braconidae) is a synovigenic, solitary, koinobiont, endoparasitoid species native to the Oriental region, where it attacks *Bactrocera* spp. larvae. This species can develop inside second- and third-instar larvae of several tephritid fruit fly species [18], subsequently emerging from the host puparium approximately fifteen days later [18,19]. It is the parasitoid species most frequently used as a fruit fly biocontrol agent in augmentative release programs worldwide [17,20,21,22]. The introduction and release of *D. longicaudata* in the Americas has, over time, contributed to the development of augmentative biological control against the Neotropical pest species *Anastrepha ludens* (Loew), *A. suspensa* (Loew), *A. obliqua* (McQuart), *A. serpentina* (Wiedemann), *A. striata* (Schiner), *A. fraterculus* (Wiedemann), and the exotic *Ceratitis capitata* (Wiedemann) [22,23,24]. Consequently, biofactories for the mass production of *D. longicaudata* have been established in different Latin American countries, including Mexico, Brazil, Peru, and Argentina [19,24,25]. An augmentative parasitoid release program requires the production of millions of parasitoids of high quality. Therefore, quality control parameters must be constantly and carefully tested throughout the parasitoid mass-rearing process. In addition, behavioral aspects of parasitoid females that affect these parameters must also be analyzed. Thus, studies have addressed mutual interference of foraging females and superparasitism in mass-reared *D. longicaudata* [24,25,26]. Self-superparasitism can be considered an advantageous strategy in which a single solitary female parasitoid lays additional eggs on a previously parasitized host to maximize her reproductive output [27]. In *D. longicaudata*, self-superparasitism appears to be an innate activity to successfully parasitize fruit fly larvae [27,28,29]. Furthermore, females of *D. longicaudata* are synovigenic [28], i.e., they continue to develop new eggs throughout their lifetime, and also adjust egg production in response to environmental variations related to host availability in order to reduce the risk of egg limitation [11].

*Diachasmimorpha longicaudata* is currently mass-produced at the fruit fly mass-rearing facility of the Moscafrut Program located in Metapa, Chiapas, Mexico. In this facility, up to 50 million pupae of this parasitoid species are produced per week. As such, this is the largest production of *D. longicaudata* worldwide [19], which facilitates the release of this parasitoid for the control of fruit fly pests in fruit-growing areas in different regions of Mexico [21]. Therefore, it is mandatory to obtain adult female parasitoids of the best quality through existing mass-rearing processes, where the age of foraging females ovipositing on hosts may be crucial to maximize rearing productivity. Determining the female ages that result in the best quality parameters of mass-reared *D. longicaudata* may offer three key benefits: (a) a reduction in the number of host larvae used in cages with females of ages with low reproductive performance, i.e., those that produce a low number of offspring or mostly male individuals; (b) saving host larvae that would remain available for the Sterile Insect Technique (SIT); and (c) an increase in the emergence rate of parasitoids with higher fitness. All this involves the improvement in the quality parameters of the parasitoids reared for a more effective suppression of the target pest. Based on the above, it would be preferable to use cages with young adult *D. longicaudata* females of less than 8 days of age in the maternal colony for the mass production of this species. This could ensure a more efficient parasitoid rearing process that would result in reduced production costs and a higher number of parasitoid females suitable for field releases. Therefore, this study aimed to evaluate the egg allocation behavior of mated *D. longicaudata* females of different ages under mass-rearing conditions, as well as to assess the fitness traits of their offspring. The results are discussed in terms of improving the efficiency of the process for the mass production of *D. longicaudata* and its use in open-field augmentative biological control programs against fruit fly pests.

## 2. Materials and Methods

### 2.1. Insect Rearing Procedures

The study was carried out at the Biological Control Laboratory (= BCL) of the fruit fly mass-rearing facility of the “Moscafrut” program, which belongs to Mexico’s Secretariat of Agriculture and Rural Development (SADER, Spanish acronym) and the Inter-American Institute for Cooperation on Agriculture (IICA, Spanish acronym), and is located in Metapa, Chiapas, southern Mexico. Parasitoids and flies, i.e., *D. longicaudata* adults and *A. ludens* larvae, respectively, were provided by the Moscafrut facility. The parasitoid *D. longicaudata* has been successfully mass-reared for over 500 generations using irradiated *A. ludens* larvae. The colony is kept in 30 × 30 × 41 cm rectangular aluminum-frame cages covered with wire mesh containing 70–80 g of parasitized puparia to support 3000 to 4000 adult parasitoids at a 2:1 female: male sex ratio. At the front of each cage, two hollow aluminum squares, covered with the same 1 mm wire mesh used for the cage walls, protrude inward through two openings (15 cm × 1.5 cm, width and height). “Cassette”-type oviposition units of 23 × 14 × 1 cm are placed inside the hollow squares. Both the upper and lower parts of the “cassettes” have organdy cloth tightly fixed to the plastic frame. Parasitoid rearing cages are grouped into 8-cage modules inside a metallic structure. Approximately 2650 third-instar *A. ludens* larvae irradiated at 45 Gy and mixed with ~30 g of larval rearing diet is placed in each unit [28]. Host larvae are irradiated to prevent the emergence of adult flies from non-parasitized larvae. The larvae were irradiated at a rate of 2.36 Gy/min using a Gammabeam 127 irradiator with a Co-60 γ-ray source (^®^Nordion, Ottawa, ON, Canada). The host irradiation treatment does not affect the quality of the emerged parasitoids [19]. Adult parasitoids are kept at 22 ± 2 °C, 70 ± 5% RH, and 12:12 h L:D, and are fed daily with crystallized bee honey and provided with water. After 2 h of exposure, host larvae are collected from the “cassettes”, separated from the diet, and placed in trays with vermiculite to enable pupation. About 15 days later, parasitized puparia are packed and shipped for field releases. Quality control is performed on each parasitized puparium batch to evaluate host larval weight, volume, and mortality, pupal weight and volume, and parasitoid emergence and viability percentages [27,29].

### 2.2. Host Exposure and Female Parasitoid Ages Evaluated

Trials were conducted at the BCL under the same controlled environmental conditions as described above. Females of *D. longicaudata* aged 5–10 days were used to evaluate egg allocation under mass-rearing conditions. Six treatments were established, corresponding to each day of female parasitoid age within the specified range. Initially, a batch of approximately two million 14-day-old parasitized *A. ludens* pupae, i.e., one day before parasitoid emergence, was distributed in groups of 6000 pupae per parasitoid cage. Parasitoid males emerged 24–48 h after placing the puparia inside the cages, while parasitoid females began to emerge two days after male emergence. The day of female emergence was considered as the first day of age. About 4000 parasitoids emerged per cage, with a sex ratio of 2.5 females per male [19]. Host larvae were exposed to females when females were 5 days old. Larvae were exposed to parasitoid females for 1.5 h per day until females were 10 days old. Parasitoids were provided with bee honey and water during the trial. About nine million *A. ludens* larvae were exposed to parasitoid females of different ages in “cassette”-type oviposition units, as described above. The overall average host-to-parasitoid ratio in each rearing cage during larval exposure was 2.6:1 (larvae:female).

### 2.3. Experimental Setup

#### 2.3.1. Parasitized Host Mortality, Superparasitism, Parasitoid Emergence and Offspring Sex Ratio

Samples of 100 *A. ludens* third-instar larvae were collected from the oviposition units exposed to female parasitoids. Three samples were taken per female age at the end of the exposure time. These samples were used to assess the parameters of parasitized host mortality, superparasitism, parasitoid emergence rate, and parasitoid offspring sex ratio. Each sample of host larvae was transferred to 4.5 × 6.5 cm (diameter × height) plastic containers with larval rearing diet. One day later, the tephritid larvae were separated from the diet by washing and then returned to the same container, to which vermiculite was added as a pupation substrate. Seventy-two hours after the larvae were exposed to female parasitoids of all tested ages, samples of 100 puparia were checked under a stereomicroscope (Stemi 305^®^Carl-Zeiss, Jenna, Germany) to determine the number of scars in each puparium due to successful oviposition or attempted oviposition. The puparia were then dissected in order to determine the number of dead parasitized hosts and count the number of parasitoid first instars per puparium, as suggested by Montoya et al. [27]. The remaining 200 puparia from each treatment were kept under laboratory conditions to assess adult emergence percentage and offspring sex ratio. Each test was replicated 10 times. Parasitoid emergence was calculated as follows: Parasitoid Emergence (%) = [No. of emerged adult parasitoids/Total No. of recovered puparia] × 100. Offspring sex ratio was determined as follows: sex ratio = number of females/number of males.

#### 2.3.2. Fitness-Related Parameters of Parasitoid Offspring

Trials were carried out to compare some fitness-related parameters of *D. longicaudata* offspring (first generation = F1) produced by 5–10-day-old female parasitoids. These parameters were survival without water and food, F1 fecundity, and offspring sex ratio of the second generation (=F2). Thirty female/male parasitoid pairs were randomly taken from each maternal age group and confined in 30 × 30 × 30 cm Plexiglas cages to assess adult survival without water and food. The number of dead individuals per sex was recorded daily. Simultaneously, a similar sample of newly emerged parasitoids from each maternal age group were placed in a similar cage with water and food (bee honey) to assess their fecundity, i.e., total number of male and female offspring (F2) produced by an average F1 female. When F1 females were 5 days old, 100 *A. ludens* third-instar larvae were exposed to them for 2 h in a 10 × 0.5 cm (diameter × height) Petri dish base covered with a piece of tricot cloth, which served as an oviposition unit. At the end of the exposure time, the host larvae were placed in 4.5 × 6.5 cm (diameter × height) plastic containers with larval diet. After one day, the larvae were separated from the diet by washing and returned to the plastic containers, which were added with moist vermiculite as a pupation substrate. The containers were kept at 26 °C and 75% RH until parasitoid emergence. Hosts were exposed to parasitoids for five consecutive days. Each test was replicated 10 times.

### 2.4. Data Analysis

Percentage of parasitoid emergence and parasitoid offspring sex ratio were compared using a one-way ANOVA. Mean differences were analyzed with Tukey’s honest significant difference (HSD) test. Parasitoid offspring survival was analyzed using a log-rank test, and parasitoid offspring fecundity was analyzed with repeated measures ANOVA. Mean differences were also analyzed by Tukey’s test. Prior to the data analysis, percentage and numerical data were Box–Cox transformed to meet the assumption of homoscedasticity. All statistical analyses were performed in JMP-SAS version 11.1.1 [30] using a significance level of *p* = 0.05.

## 3. Results

Parasitized host mortality at 72 h after exposure decreased when the host larvae were exposed to older parasitoid females (Table 1). The mortality rate was significantly higher when host larvae were exposed to 5–7-day-old females, whereas the lowest mortality occurred when hosts were exposed to 8–10-day-old females (Table 1). Superparasitism, expressed as both number of scars on the host puparium and number of parasitoid first instars, mainly decreased as female age increased (Table 1). A significantly higher number of scars and parasitoid first instars was recorded in host larvae exposed to 5-day-old parasitoid females, whereas the lowest values of both parameters were observed in hosts exposed to 10-day-old females (Table 1).

The percentage of adult emergence of *D. longicaudata* offspring was statistically similar between female age groups, with an emergence rate slightly over 50% (F_9,20_ = 0.15, *p* = 0.977) (Figure 1).

However, parasitoid offspring sex ratio (females per male) showed significant differences between female age groups (F_9,20_ = 11.18, *p* < 0.001). Females of 5 and 6 days of age produced a significantly higher number of female offspring. Parasitoid females aged 7 days produced a significantly lower number of female offspring than younger females, but a substantially higher number than older females. Parasitoid females aged 8–10 days produced the lowest number of female offspring (Figure 2).

Parasitoid female offspring that emerged from hosts parasitized by maternal females of all tested ages were able to survive for up to 11 days (Figure 3A). However, F1 daughters that emerged from host larvae exposed to younger maternal females (i.e., 5–8 days old) survived the longest, whereas those emerged from hosts exposed to 10-day-old maternal females survived the shortest time (Χ^2^
_5_ = 16.16, *p* = 0.0064) (Figure 3A). Parasitoid male offspring that emerged from hosts parasitized by maternal females of all tested ages survived for up to 10 days (Figure 3B). Similarly, to female offspring, parasitoid sons that emerged from host larvae exposed to 10-day-old maternal females survived significantly less time compared to those produced by younger mothers (Χ^2^ _5_ = 17.28, *p* = 0.004) (Figure 3B).

The interaction between sex ratio and maternal female age influenced offspring fecundity (F_5,30_ = 8.68, *p* = 0.0001). F1 parasitoid females produced by 5–9-day-old mothers exhibited significantly higher fecundity compared to those produced by 10-day-old mothers (female offspring: F_5,30_ = 4.4, *p* = 0.0039; male offspring: F_5,30_ = 2.49, *p* = 0.0053) (Figure 4). The sex ratio of F2 offspring was female-biased in all treatments, and varied from 1.6 to 2.6 females per emerged male. The number of daughters produced by 10-day-old F1 parasitoid females was significantly lower and was only similar to the number of sons produced by F1 maternal females of all ages (Figure 4). F1 mothers of 10 days of age also produced a lower number of male offspring compared to females from the other age groups, while 6-day-old mothers produced the highest number of males (Figure 4).

## 4. Discussion

The age-related egg allocation behavior of *D. longicaudata* appears to mainly influence three linked parameters under mass-rearing conditions: superparasitism, host mortality, and parasitoid offspring sex ratio. Superparasitism involves ovipositing an excessive number of eggs, and has been previously considered a strategy to suppress the immunological defenses of the attacked host [31]. In the present study, superparasitism did not have a significant effect on parasitoid emergence rate and could thus be regarded as a non-influential factor in the mass production of *D. longicaudata*. However, superparasitism has been positively related to a female-biased sex ratio [27,28], and thus younger females may produce a female-biased offspring sex ratio through increased superparasitism [27,29]. This is crucial during the mass production process of the parasitoid, as it maintains a stable reproductive rate that in turn prevents a gradual decrease in the number of offspring. Furthermore, parasitoid females play a decisive suppressive role against the target pest once they are released in the field. Mass-rearing conditions promote a high-density environment with high competition and inevitable mutual interference between foraging females [28]. This involves an initial strategy of investing resources in mass rearing to produce parasitoids whose offspring will later face high levels of intrinsic competitiveness in superparasitized hosts. Eggs and larvae of solitary parasitoids that survive in superparasitized hosts are likely to have a higher developmental capacity and therefore be better competitors [32]. Although superparasitism may seem like a waste of egg resources [33], a trade-off may occur under intrinsic competition within the host, where female eggs are more likely to develop successfully [34,35,36,37]. Differential sex mortality during development in superparasitized hosts has been suggested in the parasitoid *Eupelmus veuilleti* (Crawford), since female larvae have a higher chance of winning through larval competition [38]. The higher proportion of male offspring produced by 8–10-day-old parasitoid females in the present study may be related to a lower superparasitic activity by these females. Younger females invest more energy and resources in egg production than older females, which may lead to reduced intrinsic competition in larvae exposed to the latter. In addition, egg depletion could be another factor leading to a male-biased sex ratio in the offspring of older parasitoid females [16].

Young *D. longicaudata* females (5–7 days old) producing a higher number of female eggs could also be a result of the mass-rearing process, as it favors the selection of the most fertile and precocious females [39,40]. However, superparasitism and female-biased sex ratios have also been documented in wild populations of *D. longicaudata* [32]. In an environment with low competitive pressure, the distribution of reproductive resources may not be related to female age in *D. longicaudata*. Some studies [41,42] point out that dissimilar mean superparasitism values may be linked to different female densities, although superparasitism seems to be an intrinsic trait in *D. longicaudata,* since females foraging alone also superparasitize their larval hosts [29]. In any case, it should be emphasized that the tested parasitoids came from a colony that has been mass-produced for over 500 generations, and superparasitism has been reported to be more frequent in some parasitoid species that have been kept under mass-rearing conditions for long periods [37,43]. Thus, superparasitim plays an important role in the mass rearing of *D. longicaudata* [44]. Superparasitism also may cause increased host mortality due to excessive puncturing of the host, which mainly occurred when hosts were exposed to 5–7-day-old parasitoid females. A higher host mortality caused by parasitoid females of these ages may be attributed to an increased foraging activity that results in an overload of eggs in the host [45,46]. However, even though female age had a significant effect on host mortality, it did not affect the emergence of parasitoid offspring. Interestingly, the mean host mortality values associated with the highest oviposition activity of younger females remained around 5%, which is an allowable threshold value according to the quality parameters for the mass rearing of *D. longicaudata* [19]. An increase in this parameter would cause handling problems, since mortality values higher than 5% complicate both the process of removal of parasitized puparia from the pupation substrate and of packaging of puparia prior to release [47]. Therefore, it is advisable to continuously assess the mortality of each batch of larvae parasitized by young parasitoid females during the *D. longicaudata* mass-rearing process.

The variation in parasitoid offspring sex ratio was not only closely linked to the high superparasitism activity of young maternal *D. longicaudata* females, compared to old females, but was also highly correlated with the age of ovipositing females. Therefore, this parameter may also be influenced by physiological constraints, such as sperm depletion and aging. Both factors are related to the sex determination mechanism in haplodiploid parasitoids, whereby females can deliberately manipulate the offspring sex ratio by controlling sperm access to the eggs [48,49,50,51,52]. In the present study, we observed that the sex ratio of the offspring produced by older *D. longicaudata* females was more biased towards males. Older females laying more unfertilized haploid eggs may result from physiological constraints, such as sperm depletion or lowered sperm viability [49,53,54,55], or from the control of sperm release from the spermatheca weakening with increasing maternal age [55,56]. However, parasitoid females may deliberately change the offspring sex ratio in response to their age or life expectancy [49]. This may occur when parasitoid females have been provided only with selected large hosts, on which female eggs are normally laid, throughout their lives. Alternatively, females may begin to allocate male eggs at a very advanced age. Thus, a maternal-age-dependent offspring sex ratio may be an adaptive response [49]. The combination of the variation in the levels of superparasitism and the physiological condition of *D. longicaudata* females as a function of age may have strongly influenced the offspring sex ratio in the present study. Although there was likely an effect of maternal control on the offspring sex ratio, it could not have been related to host quality, since third-instar host larvae exposed to parasitoids during the tests were large and uniform in size (~22.2 ± 0.4 mg), according to the quality parameters for the mass rearing of *A. ludens* [27,57]. However, it can be difficult to discern whether observed offspring sex ratio patterns are due to maternal control or physiological constraints [3].

Maternal female age also influenced the survival and fecundity of parasitoid offspring. The progeny from the oldest *D. longicaudata* females (10 days old) exhibited the lowest values compared to the parasitoid offspring from the other female age groups. Decreased survival and fecundity in offspring from 10-day-old mothers may be the result of resource investment in previous oviposition events, as well as a low level of intrinsic intraspecific competition, resulting in the production of lower-quality offspring [58,59]. This is consistent with some authors who have pointed out that young females of different parasitoid species produce offspring with better competitive traits [60,61].

## 5. Conclusions

The age of *D. longicaudata* females used in adult reproduction cages is a relevant factor in the optimization of the mass production process of this parasitoid. Using production cages containing young females, i.e., 5–7 days old, may provide two main advantages: (1) enhancing the quality control parameters of parasitoids by obtaining a high proportion of female offspring with improved fitness-related parameters, such as fecundity; (2) decreasing production costs by reducing the quantity of mass-reared host larvae and rearing commodities required. In addition, production cages with older parasitoid females (≥8 days old) could be removed from the colony. These findings provide practical and useful information to improve the current method for mass rearing *D. longicaudata* at the Moscafrut Program facility. Another strategy could be to adjust the time of host exposure according to female age. The improvement in these methods is of fundamental importance, since the use of mass-produced parasitoids through augmentative releases provides an eco-friendly strategy for operational programs that include actions that contribute to the establishment of fruit fly free and low prevalence areas in Mexico and other Latin American countries, largely promoting sustainable fruit production and bolstering the global economy.

## Figures and Tables

**Figure 1 insects-16-00926-f001:**
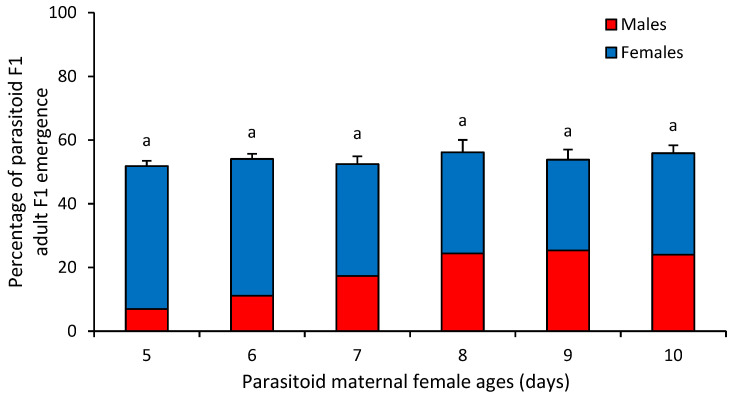
Mean (±SEM) percentage of parasitoid F1 adult emergence recorded from *Anastrepha ludens* larvae exposed to *Diachasmimorpha longicaudata* females of different ages under mass-rearing conditions. Different letters above bars indicate significant differences (Tukey’s HSD test, *p* = 0.05).

**Figure 2 insects-16-00926-f002:**
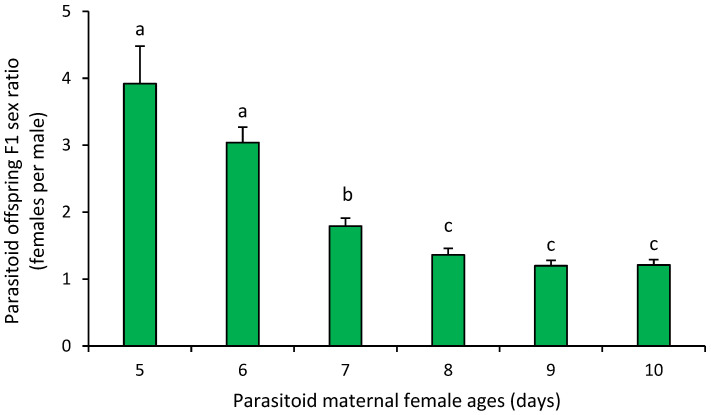
Mean (±SEM) parasitoid F1 offspring sex ratio recorded from *Anastrepha ludens* larvae exposed to *Diachasmimorpha longicaudata* females of different ages under mass-rearing conditions. Different letters above bars indicate significant differences (Tukey’s HSD test, *p* = 0.05).

**Figure 3 insects-16-00926-f003:**
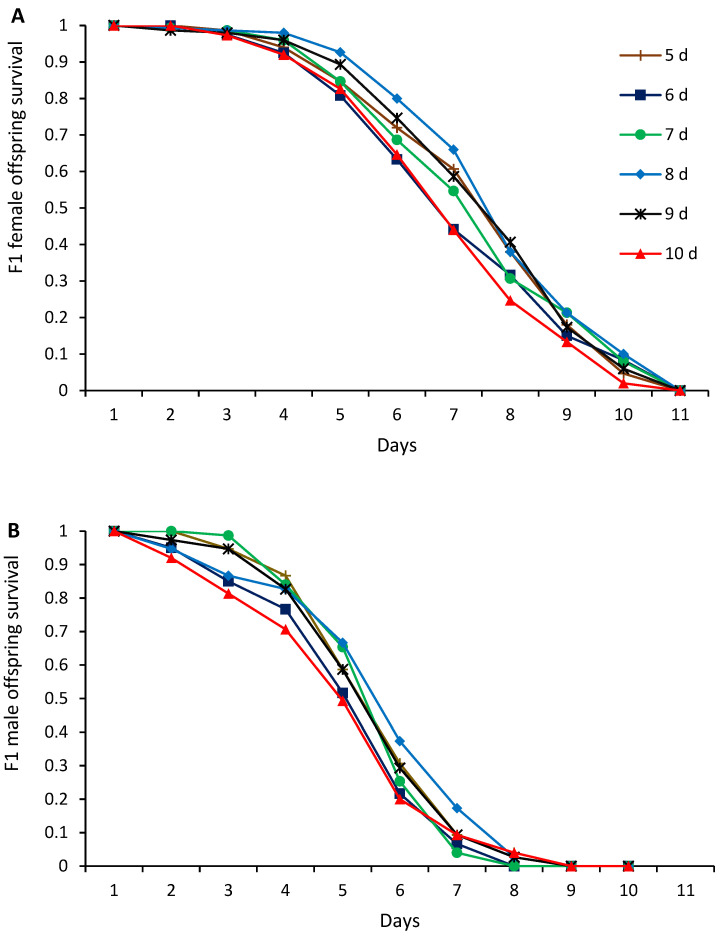
Cumulative survival curves of F1 parasitoid female (**A**) and male (**B**) offspring produced by *Diachasmimorpha longicaudata* maternal females of different ages.

**Figure 4 insects-16-00926-f004:**
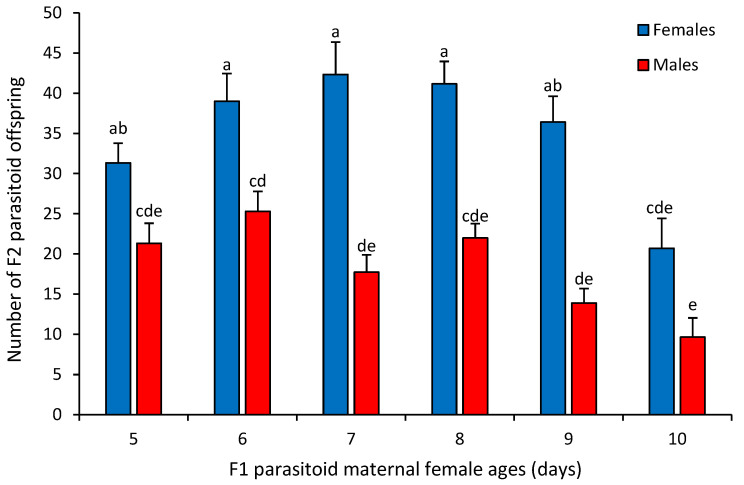
Mean (±SEM) number of F2 parasitoid female and male offspring (fecundity) produced by *Diachasmimorpha longicaudata* F1 maternal females of different ages. Different letters above bars indicate significant differences (Tukey’s HSD test, *p* = 0.05).

**Table 1 insects-16-00926-t001:** Number (mean ± SEM) of dead parasitized hosts, oviposition scars, and parasitoid first instars recorded from *Anastrepha ludens* puparia that were exposed as third-instar larvae to 5–10-day-old *Diachasmimorpha longicaudata* females under mass-rearing conditions at the Moscafrut Program facility.

Parasitoid Female Age (Days)	Parameters *		
	Dead Parasitized Hosts	Scars on Puparia	Parasitoid First Instars
5	3.0 ± 0.5 a	13.6 ± 1.1 a	5.4 ± 0.6 a
6	3.2 ± 0.5 a	8.8 ± 0.8 b	2.7 ± 0.3 b
7	3.8 ± 0.5 a	9.6 ± 1.1 ab	2.9 ± 0.3 b
8	1.7 ± 0.4 b	5.8 ± 0.4 bc	2.3 ± 0.3 bc
9	1.7 ± 0.3 b	4.0 ± 0.8 c	1.1 ± 0.3 c
10	1.5 ± 0.3 b	3.7 ± 0.6 c	1.0 ± 0.2 c
Statistical parameters	F_9,20_ = 18.97, *p* < 0.0001	F_9,20_ = 21.91, *p* < 0.0001	F_9,20_ = 18.18, *p* < 0.0001

* Means followed by different letters within the same column are significantly different (Tukey’s HSD test, *p* = 0.05).

## Data Availability

Data are available in: https://doi.org/10.5281/zenodo.15851947, accessed on 10 July 2025.

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
