# Peer review of "Egg Allocation on Anastrepha ludens Larvae by Mass-Reared Diachasmimorpha longicaudata Females"

_insects, 2025, doi:10.3390/insects16090926_

Round 1
Reviewer 1 Report
Comments and Suggestions for Authors
Review Comments to a paper
Title:
Egg Allocation on Anastrepha ludens Larvae by Mass-Reared Diachasmimorpha longicaudata Females
Manuscript code: insects-3784710
Journal name: insects
Authors in the manuscript have conceptualized a noble idea of using the appropriate parasitoid age to increase the parasitism efficiency thereby to address a couple of issues for the parasitoid release. The experiment results are straight forward and easy to understand. However, there are few points if addressed could benefit the improvement of the MS.
Please, consider italicizing the binomial names [L2, L3, L184, L210, and others]
There are many English language issues throughout the MS. Some are:
L24: Word "recurrently" seems not fitting well
L27-29: Please revise the sentence to make it much clearer.
L 44: Delete "on" from the sentence
L67-68: Better to delete "depend on the".
L117-119: Again, the sentence reads not straight. Please, revise.
L124-125: Please, revise the sentence. It is not smooth.
L132: Productively kept? I suggest REVISION.
L133: Production economic cost? Better to write something like "production cost".
L173: Sentence unclear. Please, revise.
L182: Please, revise the sentence.
L185-187: Why only the two host/parasitoid ratios [2.6:1] and why not 1.3:1 and 10.4:1?
L191-192: Please, revise English. Mind the grammar.
L205-208: Statistical representation is not ideal. Authors may insert equations to better describe their statistical methods.
L213-214: Please, revise the sentence.
Table 1: Fd,f and P-values at the end of the column would add readership of the results.
Figure 1: Error bars for male part of bars could be added.
Figure 2: What would be the result for parasitoid off-spring ratio if <5 d old maternal females were used in the study?
Based on these few comments, authors are requested to revise the manuscript intensively to meet the journal standard.
Comments on the Quality of English Language
English could be improved a lot to meet the journal standard.
Author Response
Reviewer 1
Authors in the manuscript have conceptualized a noble idea of using the appropriate parasitoid age to increase the parasitism efficiency thereby to address a couple of issues for the parasitoid release. The experiment results are straight forward and easy to understand. However, there are few points if addressed could benefit the improvement of the MS.
Please, consider italicizing the binomial names [L2, L3, L184, L210, and others]
R: Done
There are many English language issues throughout the MS. Some are:
L24: Word "recurrently" seems not fitting well
R: We changed recurrently by often
L27-29: Please revise the sentence to make it much clearer.
R: This paragraph was rewritten (see lines 25-28)
L 44: Delete "on" from the sentence
R: Done
L67-68: Better to delete "depend on the".
R: Done
L117-119: Again, the sentence reads not straight. Please, revise, and L124-125: Please, revise the sentence. It is not smooth.
R: We rewrite these lines (see lines 119-126)
L132: Productively kept? I suggest REVISION.
R: Done, see lines 131-133
L133: Production economic cost? Better to write something like "production cost".
R: Done, see line 134
L173: Sentences unclear. Please, revise.
R: We made changes, see lines 173-178
L182: Please, revise the sentence.
R: We rewrite to be clearer, lines 182-184
L185-187: Why only the two host/parasitoid ratios [2.6:1] and why not 1.3:1 and 10.4:1?
R: We worked under mass rearing conditions, thus using their general average reported during the exposition of larva to female parasitoids.
L191-192: Please, revise English. Mind the grammar.
R: Done. See lines 191, 192-194.
L205-208: Statistical representation is not ideal. Authors may insert equations to better describe their statistical methods.
R: Done, we describe the equations
L213-214: Please, revise the sentence.
R: Done
Table 1: Fd,f and P-values at the end of the column would add readership of the results.
R: Done
Figure 1: Error bars for male part of bars could be added.
R: Our intention was to show the average of total emergence. The division in females and males into the bars was only to show the general proportion of each sex, but this was not subject to statistical dimension
Figure 2: What would be the result for parasitoid off-spring ratio if <5 d old maternal females were used in the study?
R: We do not know exactly, but previous evaluations have shown that younger females (i.e., 1-4 day old) have lower fecundity, which start increasing at 5 days old.
Based on these few comments, authors are requested to revise the manuscript intensively to meet the journal standard.
R: We did it, please see all paragraphs and words in the text marked in red, written to make clearer our manuscript.
Reviewer 2 Report
Comments and Suggestions for Authors
The reported parasitoid D. longicaudata has already been used as a fruit fly biocontrol agent in augmentative release program, so the rearing techniques should be developed. However, this ms. only conduct very simply research on maternal age influence on offspring and not suitable to be concern to publish. Besides this, there are some shortcomings in the ms.
- English of the ms. is not good enough to be an article, and the author need to polish it.
- Simple summary is a summary but not an introduction, and it is too long. The authors need to reduce the sentence and only report the main result.
- Why the author only chose 5-10 d-old females to do the test but not from the 1st day aged mated female, need more explanation.
- What is “Exposure time”, exposure to what. The author should give full explanation.
- The experiment design in 2.2 should have similar sequence in the result part. The 2.2.1 should not be treated as a separate experiment, it is a basis for other tests. Or the author should give more precise subtitle to each part.
In the result part: the author only report the several parameters including parasitized host and scars on pupa and first instar larvae of parasitoid, F1 parasitoid emergence, sex ratio, and survival curves of F1 parasitoid. They are not deeply studied. Actually we can obtain the sex ratio from figure 1, so the figure 2 is not necessary.
Author Response
Reviewer 2
The reported parasitoid D. longicaudata has already been used as a fruit fly biocontrol agent in augmentative release program, so the rearing techniques should be developed. However, this ms. only conduct very simply research on maternal age influence on offspring and not suitable to be concern to publish. Besides this, there are some shortcomings in the ms.
R: The reviewer is right, the mass rearing of this parasitoid species has been developed in the past, but for the same reason, nowadays it is also under continuous scrutiny to improve their rearing efficiency, looking for a higher production, better adult females’ quality and lower production costs. The research on maternal age under mass rearing conditions is a high demandant labor that requires the simultaneous evaluation of several key parameters to maintain or improve the demanded quality of adult parasitoids. We believe that this work provides new and valuable evidence about the role of maternal age to improve the rearing efficiency of this parasitoid species.
English of the ms. is not good enough to be an article, and the author need to polish it.
R: The English was reviewed by and expert English speaker. We hope now to reach the standards of the Insects journal.
Simple summary is a summary but not an introduction, and it is too long. The authors need to reduce the sentence and only report the main result.
R: done, we shortened this part.
Why the author only chose 5-10 d-old females to do the test but not from the 1st day aged mated female, need more explanation.
R: we worked under the mass rearing conditions, and these ages (5-10 d-old females have been selected because contain females with higher fecundity. The younger females (i.e., 1-4 day old) offer a low fecundity affecting thus the rearing efficiency.
What is “Exposure time”, exposure to what. The author should give full explanation.
R: We changed to “the period of larval exposition to parasitoids”
The experiment design in 2.2 should have similar sequence in the result part. The 2.2.1 should not be treated as a separate experiment, it is a basis for other tests. Or the author should give more precise subtitle to each part.
R: We changed the names of each section. “Experimental” setup was moved for description of experiments. Two initial sections were separated to explain better the general conditions of the experiments.
In the result part: the author only report the several parameters including parasitized host and scars on pupa and first instar larvae of parasitoid, F1 parasitoid emergence, sex ratio, and survival curves of F1 parasitoid. They are not deeply studied. Actually we can obtain the sex ratio from figure 1, so the figure 2 is not necessary.
R: This work was designed to explain the effects of the age parasitoid female in relationship with the different parameters obtained in the mass production. A more complete study would be interesting (for example analyze all female ages), but nowadays exist different studies reporting this, and, for mass rearing production, the youngest females (1-4 day old) are not interesting for reproduction in mass rearing conditions, because of their lower fecundity.
Regarding the figures, in the number 1 we want to show just the total adult emergence and its relationship with the sex proportion. In the Fig. 2, the objective was to focus only on the sex-ratio, doing emphasis in the proportion of females.
Reviewer 3 Report
Comments and Suggestions for Authors
I found the manuscript # Insects-3784710 very interesting and well written. Most of my comments (in the PDF) are just editorial editing mistakes.
However, I suggested to improve a little in the introduction (line 94) the biology of the parasitoids, even if some of the questions are answered later in the material and methods chapter.
Finally, during the discussion, I found intriguing the aspect of the super parasitism, especially done by the young females..and I'm wondering and I am asking to the authors if they were considering if this peculiar behavior is more frequent in tephritid larvae that have been previously parasitized by old (poorly performant) D. longicaudata females.

Author Response
Reviewer 3
I found the manuscript # Insects-3784710 very interesting and well written. Most of my comments (in the PDF) are just editorial editing mistakes.
However, I suggested to improve a little in the introduction (line 94) the biology of the parasitoids, even if some of the questions are answered later in the material and methods chapter.
R: Done, see lines 93-97.
Finally, during the discussion, I found intriguing the aspect of the super parasitism, especially done by the young females..and I'm wondering and I am asking to the authors if they were considering if this peculiar behavior is more frequent in tephritid larvae that have been previously parasitized by old (poorly performant) D. longicaudata females.
R: Thank you for your comment, it is very interesting. There are some reports on the discrimination ability of this parasitoid species, describing it as innate but increasing with the learning. However, in this study performed under mass rearing conditions this scenario (young female ovipositing in larvae previously attacked by old females) is not possible, because all females are maintained in cages with conspecifics of the same age. Under field conditions can occur that young females use the larva previously oviposited by old females, but if there are some preferences to do this ratter to attack larvae oviposited by young females, needs to be investigated.
Round 2
Reviewer 1 Report
Comments and Suggestions for Authors
Authors have improved their manuscript significantly. However, some of the issues are yet to be addressed, especially those raised against statistical values. Following issues are expected to be addressed:
Table 1: How df=5.97? Since, the authors are using ten female ages as treatments replicated thrice, they are expected to write statistical values as F9, 20 = 18.97, P < 0.0001 for the first column.
Similar comment for column 2 and 3 of the Table 1.
L209-210: Better to write formula like:
Parasitoid Emergence (%) = [No. of emerged adult parasitoids/ Total No. of recovered puparia]*100.
Fig 1: Statistical explanations for Fig.1 and Fig. 2 are also needed to improve. Eg F9,20=0.15, P = 0.977.
L317-318, L320-321.: Please, revise the statistical values with correct writing style as explained above.
Author Response
Table 1: How df=5.97? Since, the authors are using ten female ages as treatments replicated thrice, they are expected to write statistical values as F9, 20 = 18.97, P < 0.0001 for the first column.
Done
Similar comment for column 2 and 3 of the Table 1.
Done
L209-210: Better to write formula like:
Parasitoid Emergence (%) = [No. of emerged adult parasitoids/ Total No. of recovered puparia]*100.
Done
Fig 1: Statistical explanations for Fig.1 and Fig. 2 are also needed to improve. Eg F9,20=0.15, P = 0.977.
Done
L317-318, L320-321.: Please, revise the statistical values with correct writing style as explained above.
These values were corrected according the comments
Reviewer 2 Report
Comments and Suggestions for Authors
After revision, the ms. improve a lots and could be accepted.
Author Response
Thanks for your help.